# A Multi-Modal and Multi-Atlas Integrated Framework for Identification of Mild Cognitive Impairment

**DOI:** 10.3390/brainsci12060751

**Published:** 2022-06-08

**Authors:** Zhuqing Long, Jie Li, Haitao Liao, Li Deng, Yukeng Du, Jianghua Fan, Xiaofeng Li, Jichang Miao, Shuang Qiu, Chaojie Long, Bin Jing

**Affiliations:** 1Medical Apparatus and Equipment Deployment, Hunan Children’s Hospital, Changsha 410007, China; longzhuqing16@sina.com (Z.L.); jielilove08@sina.com (J.L.); hticf_fr2014@outlook.com (H.L.); snow407@163.com (Y.D.); ss123815779@sina.com (S.Q.); 2School of Biomedical Engineering, Capital Medical University, Beijing 100069, China; 3Department of Data Assessment and Examination, Hunan Children’s Hospital, Changsha 410007, China; dengli0305@sina.com; 4Department of Pediatric Emergency Center, Emergency Generally Department I, Hunan Children’s Hospital, Changsha 410007, China; fjhlsl_2008@163.com; 5Hunan Guangxiu Hospital, Hunan Normal University, Changsha 410006, China; lx_xf0521@163.com; 6Department of Medical Devices, Nanfang Hospital, Guangzhou 510515, China; lzjxh@i.smu.edu.cn

**Keywords:** multi-modal neuroimaging, appropriate atlas, mild cognitive impairment, gray matter volume, Hurst exponent, support vector machine

## Abstract

Background: Multi-modal neuroimaging with appropriate atlas is vital for effectively differentiating mild cognitive impairment (MCI) from healthy controls (HC). Methods: The resting-state functional magnetic resonance imaging (rs-fMRI) and structural MRI (sMRI) of 69 MCI patients and 61 HC subjects were collected. Then, the gray matter volumes obtained from the sMRI and Hurst exponent (HE) values calculated from rs-fMRI data in the Automated Anatomical Labeling (AAL-90), Brainnetome (BN-246), Harvard–Oxford (HOA-112) and AAL3-170 atlases were extracted, respectively. Next, these characteristics were selected with a minimal redundancy maximal relevance algorithm and a sequential feature collection method in single or multi-modalities, and only the optimal features were retained after this procedure. Lastly, the retained characteristics were served as the input features for the support vector machine (SVM)-based method to classify MCI patients, and the performance was estimated with a leave-one-out cross-validation (LOOCV). Results: Our proposed method obtained the best 92.00% accuracy, 94.92% specificity and 89.39% sensitivity with the sMRI in AAL-90 and the fMRI in HOA-112 atlas, which was much better than using the single-modal or single-atlas features. Conclusion: The results demonstrated that the multi-modal and multi-atlas integrated method could effectively recognize MCI patients, which could be extended into various neurological and neuropsychiatric diseases.

## 1. Introduction

Mild cognitive impairment (MCI), generally representing a transition stage between normal aging and Alzheimer’s disease (AD) [1,2], is clinically characterized by intellectual deficits, memory complaints and other reduced cognitive functions [3,4]. Overall, MCI patients progress to AD at an annual rate of 15–26% [5], and half of them will convert to AD within 3–5 years [6]. As therapeutic treatments become available, objective and valid biomarkers, which could serve as in vivo surrogates for pathological changes in MCI patients, are desperately needed because efficient treatments need early initiations before irreversible brain damage occurs [7].

Multi-modal neuroimaging techniques, such as resting-state functional magnetic resonance imaging (rs-fMRI) and structural MRI (sMRI), have been widely utilized to characterize abnormalities in MCI and AD patients [8,9], and the detected abnormal regions were primarily located in the posterior cingulate gyrus, hippocampus and amygdala, etc. [10,11,12]. In addition, several studies have tried to combine multi-modal neuroimaging data together to distinguish MCI or AD patients from healthy controls (HC) because different imaging modalities provide complementary information to each other in comparison to a single modality [13,14,15]. However, the multi-modal integration results for MCI or AD identification were not consistent. Some previous studies demonstrated that integrated multi-modal data improved the classification performance in differentiating MCI or AD patients [16,17], while another study concluded the integration did not promote the classification accuracy [18]. These discrepancies indicate that the multi-modal integration based on MCI or AD discrimination needs to be further investigated.

Especially, appropriate brain parcellation is vital to quantitatively detect the functional and structural abnormalities in MCI and AD patients, but there is no golden standard atlas for each modality in MCI or AD classification. Currently, the Automated Anatomical Labeling (AAL-90) atlas is a popular one [19,20,21], however, it is not refined enough in some brain regions. Therefore, some more detailed atlases have been proposed, such as the Harvard–Oxford atlas (HOA-112), the Brainnetome atlas (BN-246), and the newly proposed AAL3-170 atlas [22,23,24]. Different atlases bring about a multi-scale perspective of the whole brain, which may shed light on the multi-modal integration.

Further, sMRI provides morphological information about the macroscopic brain tissues, which have been widely adopted to reveal brain atrophy underlying MCI or AD patients [7]. In contrast, rs-fMRI offers functional signal characteristics (e.g., fluctuation and coupling), which have been popularly utilized for the diagnosis of MCI and AD patients [25,26]. Early studies reflect that the blood oxygenation level-dependent (BOLD) signal in the brain displays scale-free or fractal-like dynamics [27,28]. The fractal-like dynamics stand for the phenomenon that there is self-similarity in the time course of the fMRI signal. For example, a voxel with a Hurst exponent (HE) larger than 0.5 indicates the positively correlated BOLD series, i.e., the changing trend in future time points is similar to previous time points. Currently, the HE index has been applied to investigate the characteristics of rs-fMRI signals in autism disorder, normal and pathological aging, major depressive disorder, AD and individual traits [27,28,29,30,31]. In our previous studies, we found that the performance of the HE index and the gray matter volume in MCI classification were both dependent on the brain atlas [6,32]; however, it is still unknown whether the combination of different atlases could further improve the recognition performance of MCI patients.

In this study, we proposed a multi-modal and multi-atlas integrated framework to identify MCI from HC subjects. In detail, the mean gray matter volumes obtained from the sMRI and the mean HE values calculated from rs-fMRI in the AAL-90, BN-246, AAL3-170 and HOA-112 atlases were extracted, respectively. Then, these candidate features were selected with a minimal redundancy maximal relevance (MRMR) algorithm and a sequential feature collection (SFC) method, and only the remaining optimal features were served as the input features to construct a support vector machine (SVM)-based modal to identify MCI form HC subjects. Lastly, the classification performance was compared between the proposed method and several other benchmark models.

## 2. Materials and Methods

### 2.1. Participants

A total of 69 MCI and 61 HC subjects were enrolled in this study, and all subjects did not take any medications that may have interfered with cognitive functions before the scan. All patients were collected from the memory clinic of the neurology department in Nanfang Hospital, which is affiliated with Southern Medical University, and all HC subjects were collected from the local community by posting advertisements. This study was in accordance with the medical research ethics committee of Nanfang Hospital, and the informed written consent from all participants was obtained following the rules of the Declaration of Helsinki. All subjects were right-handed, and the subjects of the two groups matched well in sex, age and years of education. Before they took part in this study, all subjects underwent physical and psychological examinations, and the cognitive functions of all subjects were assessed with a standard clinical evaluation, including the Clinical Dementia Rate (CDR), the Mini-Mental State Examination (MMSE) and the Auditory Verbal Learning Test (AVLT). A total of 3 MCI and 2 HC subjects were discarded for excessive head motion, and the detailed neuropsychological and demographic characteristics of the remaining subjects are shown in Table 1. All subjects were diagnosed by two experienced experts with the following criteria.

MCI criteria: (1) subjective memory complaints, verified by themselves or their relatives; (2) normal or near-normal performance of cognitive functions; (3) normal or near-normal activities of daily living; (4) a CDR score of 0.5; (5) not meeting the dementia criteria according to the DSM-IV (Diagnostic and Statistical Manual of Mental Disorders, 4th edition, revised); (6) a cutoff point of AVLT-delay recall: 6 [33]; (7) a threshold of MMSE score: 19 (no formal education), 22 (1 to 6 years of education), 26 (7 or more years of education) [33].

HC criteria: (1) a CDR score of 0; (2) normal cognitive and physical status; (3) without memory complaints; (4) normal activities of daily living.

Exclusion criteria for all subjects were listed as: (1) no other nervous system diseases that result in cognitive impairments, such as brain tumors, major depressive disorder and Parkinson’s disease; (2) no systemic diseases that intervene with cognitive functions, such as severe anemia and syphilis; (3) no history of stroke and alcohol dependence; (4) and no visible vascular lesions on the sMRI.

### 2.2. Data Acquisition

All data were acquired on a 3.0 Tesla Siemens scanner with an 8-channel radio frequency coil at Nanfang Hospital. Comfortable foam paddings and a headphone were simultaneously used to minimize head motion and reduce the scanner noise during the scan. All participants were told to keep their eyes closed and their minds relaxed, not to fall asleep and not to move their heads as much as possible. Rs-fMRI images were collected with an echo-planar imaging (EPI) sequence by using the following parameters: repetition time (TR) = 2000 ms, echo time (TE) = 40 ms, flip angle (FA) = 90°, matrix size = 64 × 64, field of view (FOV) = 240 × 240 cm^2^, thickness = 4 mm, voxel size = 3.75 × 3.75 × 4 mm^3^. A total of 239 volumes were obtained for all subjects within 478 s. Structural images were collected utilizing a magnetization-prepared rapid gradient echo (MPRAGE) T1-weighted sequence with the following parameters: TR = 1900 ms, TE = 2.2 ms, inversion time = 900 ms, FA = 9°, matrix size = 256 × 256, number of slices = 176, thickness = 1 mm, voxel size = 1 × 1 × 1 mm^3^.

### 2.3. Data Preprocessing

#### 2.3.1. fMRI

Data preprocessing for the fMRI data was performed via Statistical Parametric Mapping (SPM8, http://www.fil.ion.ucl.ac.uk/spm, accessed on 4 June 2022). The first 10 functional images for all subjects were excluded from analysis, and the remaining 229 volumes were corrected for different acquisition times between slices. Then, all volumes were realigned to the first image to compensate for head movement effects. A total of 5 subjects, including 3 MCI and 2 HC, were discarded due to excessive head motion (2 mm and 2° in all directions). To improve the spatial normalization accuracy of the fMRI data, the structural images for all subjects were first co-registered to the functional data, and the co-registered sMRI data were segmented and then normalized to the standard Montreal Neurological Institute (MNI) space. The realigned functional images were normalized to the MNI space by utilizing the parameters obtained from the previous step and then resampled into a voxel of 3 × 3 × 3 mm^3^. Several spurious covariates, including the 6 head-motion parameters, the average signals in white matter and cerebrospinal fluid, and the linear drift were regressed out from the normalized fMRI data. Lastly, all the regressed images were filtered with a temporal filter (0.01–0.10 Hz) to reduce high-frequency noise and low-frequency drift and were smoothed with a 4 mm full width at half maximum.

#### 2.3.2. sMRI

All sMRI images were carried out with the VBM8 toolbox implemented in SPM8, with the following procedures. Firstly, all sMRI images, checked by two experienced neuroradiologists with no significant artifact and abnormality, were segmented into white matter, gray matter and cerebrospinal fluid by the ‘New Segment’ tool in the SPM. Then, all these segmented images were normalized into the Montreal Neurological Institute (MNI) space by the diffeomorphic anatomic registration through the exponentiated lie (DARTEL) algorithm, and then the Jacobian matrices were utilized to modulate the normalized images to preserve the tissue volume information. Lastly, all these modulated data were smoothed with an 8 mm full width at half the maximum Gaussian kernel.

### 2.4. Feature Extraction under Four Atlases

The range scaled (R/S) method was utilized to calculate the HE values at a voxel level, and the detailed principles for the calculation of the HE index were described in previous studies [27,34]. In this paper, the averaged HE values in every region of interest (ROI) of the AAL-90, BN-246, AAL3-170 and HOA-112 atlases were extracted, respectively, as the candidate features to identify the MCI from HC subjects. The AAL-90 atlas partitions the whole cerebral cortex into 90 regions (without cerebellum), while the BN-246 atlas contains 210 cortical and 36 subcortical brain regions. The AAL3-170 atlas (https://www.oxcns.org/aal3.html, accessed on 4 June 2022) is an improved version of the AAL-90 atlas that partitions the whole brain into 166 ROIs. Moreover, two small regions of the AAL3 atlas were not defined (numbers 133–134), as the original voxel size of 1 × 1 × 1 mm^3^ was resampled into 3 × 3 × 3 mm^3^, and the cerebellum regions (numbers 95–120) were excluded; therefore, the remained number of regions of the AAL3-170 atlas was 138. The HOA-112 atlas partitions the brain into 112 brain regions, but the brain stems (numbers 97–98) in the HOA-112 atlas were excluded for subsequent analysis. Based on the above-mentioned four atlases, the processed sMRI images were also employed to extract the volume in every ROI of these four atlases, respectively, by using the following Matlab code (http://www.cs.ucl.ac.uk/staff/G.Ridgway/vbm/get_totals.m, accessed on 4 June 2022). The above-mentioned four atlases used for the calculation of functional and structural features are shown in Figure 1.

### 2.5. Feature Selection

Considering that some features are irrelevant or redundant for MCI identification, a feature selection algorithm is essentially needed to obtain the optimal features for classification. Prior studies have indicated that correctly reducing the number of features can not only speed up computation but also improve the classification performance [19,35]. Therefore, the MRMR method, in combination with the SFC algorithm, was utilized for obtaining a subset of discriminative features. In detail, the MRMR score for a feature set is defined as:(1)$$MRMR=MAX_{s}\{\frac{1}{\left|s\right|}\sum_{f_{i}\in s}I(f_{i},c)-\frac{1}{|s{|}^{2}}\sum_{f_{i}\in s}I(f_{i},f_{j})\}$$
where the relevance between the feature set *S* and *K* classes *C* = {*C*_1_*, C*_2_*, C*_3_*, …, C_k_*} is calculated by the mean values of mutual information between the individual feature *f_i_* and *C*, and the redundancy of all features in the feature set *S* is the mean value of mutual information between features *f_i_* and *f_j_*. The top 50 features extracted by the MRMR method were then utilized for the SFC algorithm to select the optimal subset of features [36]. In detail, the first feature was selected as the starting point, and the first 2 features of the extracted 50 features were used to compute the classification performance. Then, the first 3 features were utilized for computing the classification performance, and the procedure was continued until all 50 features were used. After this loop, the first feature was eliminated and the second feature served as the starting point, and the classification process was repeatedly performed with the feature numbers, ranging from 2 to 49. The starting point was then circularly set from the 3rd feature to the 49th feature to repeat a similar process. Lastly, the optimal subset of features was determined by comparing the classification performance of all the subsets.

### 2.6. SVM Based Classification

The SVM method aims to seek the optimal class-separating hyper-plane with the maximum margin in the feature space [37]. In this paper, the LibSVM package (http://www.csie.ntu.edu.tw/~cjlin/libsvm, accessed on 4 June 2022) integrated into Matlab was utilized for SVM implementation, and the radial basis function (RBF), which could deal with the nonlinear relationships between the feature vectors and the class labels, was served as the kernel function in SVM. In addition, a grid search method was adopted to optimize two parameters of SVM: the *C*, adjusting the importance of error separation, and the γ, representing the width of the RBF kernel function with the adjusting range of $C=2^{-8},2^{-7.5},\ldots ,2^{8}$ and $\gamma =2^{-8},2^{-7.5},\ldots ,2^{8}$. These two parameters were optimized by an internal LOOCV loop that was only carried out on the training data, and an external LOOCV loop was performed to estimate the classification performance of accuracy, sensitivity and specificity, which represents the correct discrimination rate of all samples, MCI patients and HC subjects, respectively. It is worth noting, however, that the parameter optimization and feature selection were only carried out on the training data, and the classification performances of these optimally combined features selected by the SFC algorithm were deemed as the final results.

In this paper, four different classification strategies were adopted and compared, including single-modality with single-atlas, single-modality with multi-atlas, multi-modality with single-atlas and multi-modality with multi-atlas. First, single-modality with single-atlas models were, respectively, constructed by applying every atlas to GMV or HE to identify MCI patients. After that, multi-atlas bagging on single-modality models was constructed by using the optimal three atlases from each modality to form a major voting model. Then, the multi-modality with single-atlas models was created by applying every atlas to multi-modality data, respectively. Lastly, the optimal two atlases from each modality were selected to generate four kinds of multi-modal and multi-atlas integration frameworks for MCI identification. Notably, not all atlases were used here in order to decrease the computational burden.

## 3. Results

### 3.1. Classification Performance of Different Models

By applying the proposed classification method to identify MCI patients from HC subjects, our proposed multi-modal and multi-atlas integration method obtained a best accuracy of 92.00%, a specificity of 94.92% and a sensitivity of 89.39% when using the structural data in the AAL-90 atlas and the functional data in the HOA-112 atlas. In contrast, when single-modal and single-atlas features were used, rs-fMRI obtained a best accuracy of 87.20%, a specificity of 86.44% and a sensitivity of 87.88% with the HOA-112 atlas, and sMRI achieved a best accuracy of 84.8%, and a specificity of 88.14% and a sensitivity of 81.82% with the AAL-90 atlas. When using single-modality data with multi-atlas, the bagging results of sMRI and rs-fMRI achieved an accuracy of 86.40% and 88.80%, respectively. Furthermore, when using multi-modality data with a single atlas, the best performance was received by the HOA-112 atlas with an accuracy of 88.0%. The comparisons of classification performance in different models are summarized in Table 2. In addition, the receiver operating characteristics (ROC) curves of single-modality with the single-atlas model and multi-modality with the multi-atlas model are shown in Figure 2, and the best area under curve (AUC) values of the corresponding models were 0.9081 and 0.9502, respectively, indicating a powerful discrimination ability of our proposed method. Lastly, the best classification results under a different number of features ranging from 2 to 50 are shown in Figure 3.

### 3.2. Between-Group Differences in HE Index and Gray Matter Volume

Figure 4 displayed the abnormal brain regions that demonstrated the most discriminative powers for identifying MCI patients from HC subjects in a single atlas or different combinations of multi-atlases. Overall, the structural gray matter volume abnormalities were predominately involved in the bilateral posterior cingulate gyrus, bilateral amygdala, left inferior frontal gyrus, right hippocampus, left basal ganglia and left putamen. The functional HE abnormalities were mainly located in the bilateral hippocampus, bilateral inferior frontal gyrus, bilateral thalamus, left fusiform, left posterior cingulate gyrus and left putamen. In addition, the weighted contributions of these most discriminative features in the single-modal and multi-modal models are shown in Figure 5.

## 4. Discussion

In this study, we proposed a multi-modal and multi-atlas integrated framework to identify MCI patients from HC subjects, and compared the performance with three other kinds of single-modal or single-atlas models. Our results found that the classification performance of the proposed method was superior to these benchmark models, and obtained the best accuracy of 92% when applying the AAL-90 on GMV and the HOA-112 atlas on HE, indicating these multi-modal data were effectively fused in the MCI classification. Thus, this proposed method is effective in detecting complementary and comprehensive information from multi-modal data for MCI classification.

To improve the classification performance, three key elements were taken for the proposed integration method to identify MCI patients. First, considering multi-modal neuroimaging data can provide complementary information compared to a single modality, thus sMRI and fMRI data were both utilized for MCI discrimination, and our results validated that the combined information can enhance the classification performance. Second, some studies suggested that correctly selecting the optimal features could not only speed up the computation but also improve the classification performance [19,32], thus the MRMR method, together with the SFC algorithm, was adopted for feature selection, and the discrimination performance was significantly enhanced in comparison to the model without feature selection. In fact, we attempted the proposed classification method for all features without feature selection, and the accuracy rates were 62.40%, 59.20%, 59.20% and 60.80% using single functional data and 57.60%, 59.20%, 57.60% and 64.80% by using single structural data in the AAL-90, BN-246, AAL3-170 and HOA-112 atlases, respectively, which were significantly lower than those after feature selection. It is worth noting that the feature selection was only carried out on the training data, which can avoid the over-fitting of the classifier. Third, the RBF kernel function, which can handle the condition when the relationships between labels and features are nonlinear, and the grid search method, which has a high learning accuracy and can be implemented with parallel processing, were simultaneously utilized for MCI classification, which also had an important impact on classification performance. In addition, we have tested the linear kernel function to replace the RBF kernel in the single-modal models, and the recognition rates were 80.00%, 76.80%, 76.00% and 76.80% using single structural data and 74.40%, 76.00%, 77.60% and 81.60% using single functional data in the AAL-90, BN-246, AAL3-170 atlases and HOA-112 atlas, respectively, which were lower than those with the RBF kernel function. Taken together, the proposed method is more effective in identifying MCI patients from HC subjects.

In this paper, the overlapping abnormal brain regions in both the gray matter volume and the functional HE characteristic were involved in the left posterior cingulate gyrus, left inferior frontal gyrus, right hippocampus and left putamen. All these abnormal regions were consistent with prior studies that analyzed the functional or structural data of MCI patients with conventional statistical analyses [2,19,38]. The posterior cingulate gyrus and hippocampus belong to the default mode network (DMN), supporting the aberrant behaviors of DMN in MCI patients, which was consistent with many previous MCI or AD studies [39,40]. The inferior frontal gyrus has been detected with a significant amplitude of low-frequency fluctuations (ALFF) abnormality [41], and the putamen was found with significant atrophy in MCI patients [42]. These consistent findings suggested that the abnormalities in these regions were associated with the mechanisms underlying AD and MCI patients. Moreover, some discrepancies in the detected abnormal regions between structural data and functional data were also found, such as the gray matter atrophy in the amygdala and basal ganglia, and the HE abnormalities in the thalamus and fusiform. The main reason for these discrepancies may be attributed to the specificity of sMRI and fMRI. The gray matter volume obtained from the sMRI data reflects the morphological information, and the abnormalities in the amygdala and basal ganglia were consistent with prior MCI or AD studies [42,43]. The HE index acquired from the fMRI data reflects the persistent behavior of brain activities, and it has been even proposed as a measure of online information-processing efficiency: higher HE values are related to longer memory dynamics, higher temporal redundancy and less freedom to vary [28]. The HE abnormalities in the thalamus and fusiform reflect that the persistent pattern in these regions was changed, which may provide a unique perspective to understand the functional alterations in MCI patients.

Different brain parcellation schemes may generate ROIs with distinct sizes and locations, which results in a unique feature representation and therefore influences the classification performance. The AAL-90 atlas is created based on 27 high-resolution T1-weighted images of a young man, and the initial aim of this template was to offer a standard anatomical reference for fMRI data. However, the borders of every ROI in the atlas were defined using sulcal landmarks but with poor consistency to the cytoarchitectonic borders [44,45], resulting in variable sulcal and gyral patterns [45]. The HOA-112 atlas incorporates information on sulcal and gyral geometry [46], which may better reflect the individual variability. The BN-246 atlas is a probabilistic atlas generated from 40 MRI data of healthy subjects, which is created by identifying sub-regions that were maximally different from each other and maximally homogeneous internally with the local structural connectivity [23]. Our single-modality with single-atlas models revealed that different atlases obviously affected the MCI classification, implying future studies should pay more attention to the choice of brain atlas in related studies. Moreover, the bagging strategy did not improve the classification performance significantly, indicating that multi-atlas fusion in single-modal data may be limited by the similarity of extracted information from these atlases. Furthermore, multi-modal data with a single atlas also did not promote the classification accuracy, which may be biased by the atlas dependent attribute for multi-modal imaging data.

Several limitations need to be mentioned. First, all the selected atlases in the study exclude the cerebellum and brain stem, which may also provide some contribution to MCI discrimination. Second, some other atlases existed in the neuroimaging studies nowadays, and these atlases can also be used for differentiating MCI patients. Third, considering the samples utilized in this work are not very large, the obtained classifier may not be robust enough. In the next step, we will apply our method to a large dataset to further validate the classification performance. Fourth, this study lacks information on amyloid-beta, tau deposition and APOE genotype, which can also affect brain structure and function.

## 5. Conclusions

In this paper, we proposed a multi-atlas and multi-modal integrated framework to identify MCI in HC subjects. The results demonstrated its obvious superiority in comparison to other single-modality or single-atlas models, which can be used to improve the early diagnosis of MCI and can be extended into other neurological and neuropsychiatric disease classifications.

## Figures and Tables

**Figure 1 brainsci-12-00751-f001:**
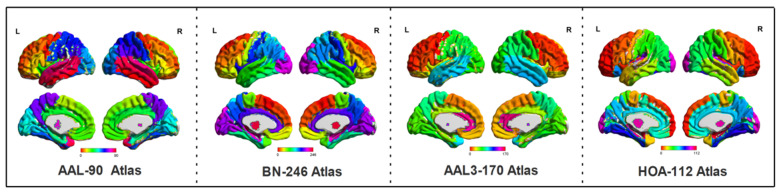
The four atlases, including AAL-90, BN-246, AAL3-170 and HOA-112.

**Figure 2 brainsci-12-00751-f002:**
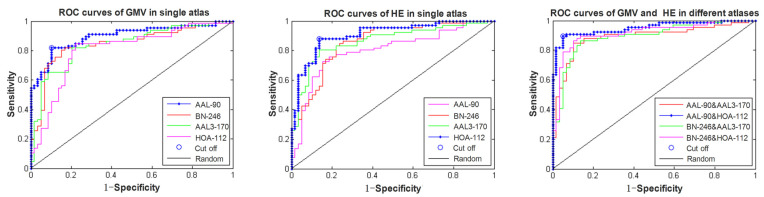
The ROC curves of the single-modality and multi-modality models.

**Figure 3 brainsci-12-00751-f003:**
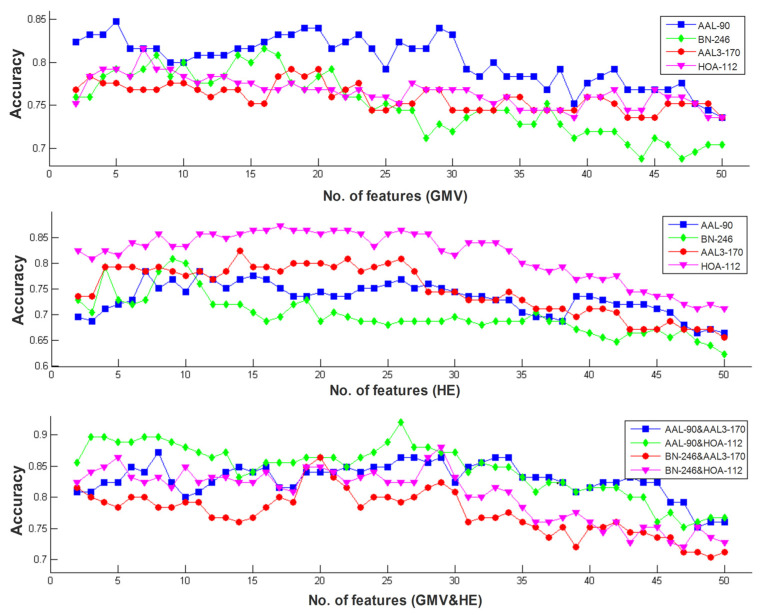
The best classification results with a different number of features ranging from 2 to 50 in single-modal and multi-modal data.

**Figure 4 brainsci-12-00751-f004:**
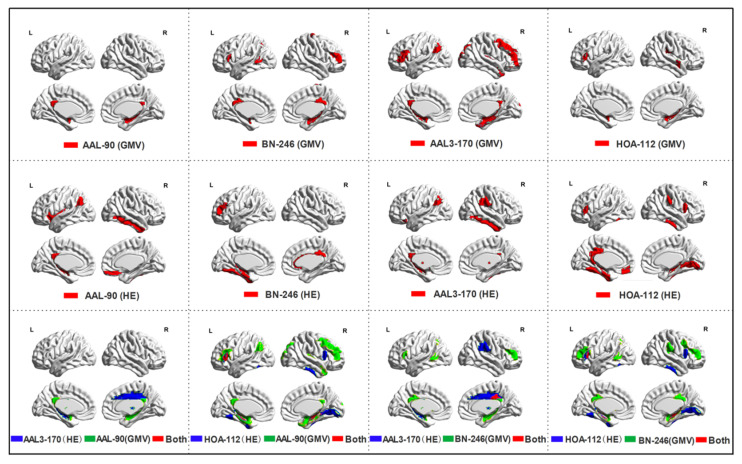
The most discriminative features in single-atlas or multi-atlas models.

**Figure 5 brainsci-12-00751-f005:**
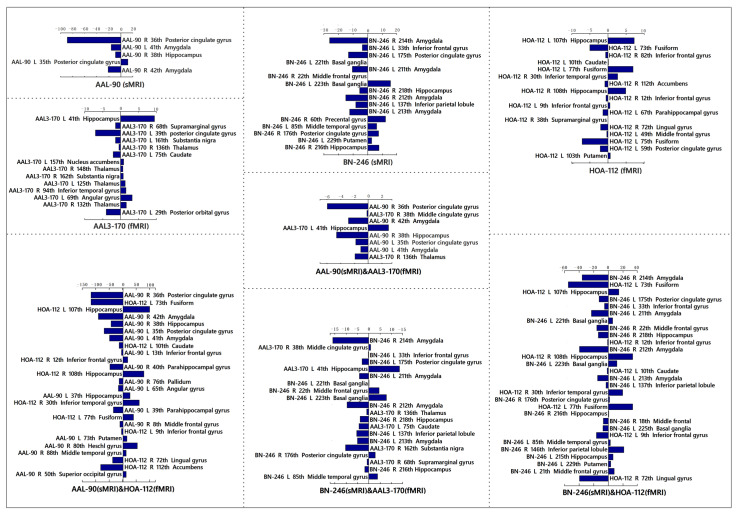
The weighted contributions of these most discriminative features in single-modal or multi-modal models.

**Table 1 brainsci-12-00751-t001:** Participants’ demographic and clinical characteristics.

Characteristics	MCI	HC	*p*-values
Gender (M/F)	66 (35/31)	59 (28/31)	0.53 ^#^
Age (years)	67.20 ± 7.22	65.22 ± 7.36	0.13 *
Education (years)	9.83 ± 4.22	10.01 ± 4.29	0.81 *
CDR	0.5	0	0 *
MMSE	23.47 ± 2.71	27.37 ± 3.17	<0.001 *
AVLT-immediate recall	7.12 ± 3.49	11.58 ± 2.25	<0.001 *
AVLT-delay recall	3.67 ± 2.85	9.80 ± 2.80	<0.001 *
AVLT-recognition	8.01 ± 2.56	12.95 ± 2.97	<0.001 *

Values are mean ± S.D unless the S.D was not calculated; M, male; F, female; ^#^ The *p*-value was obtained by Chi-square test; * The *p*-values were obtained by the two-tailed two-sample *t*-test.

**Table 2 brainsci-12-00751-t002:** The MCI classification performance in different models.

Modality	Atlases	No. Selected Features	Accuracy	Specificity	Sensitivity	AUC Values
sMRI	AAL-90	5	84.80%	88.14%	81.82%	0.8970
	AAL3-170	18	79.20%	79.66%	78.79%	0.8405
	BN-246	16	81.60%	81.36%	81.82%	0.8451
	HOA-112	7	81.60%	77.97%	84.85%	0.8046
	Bagging	28	86.40%	84.75%	87.88%	-
fMRI	AAL-90	7	78.40%	75.76%	81.36%	0.8007
	AAL3-170	14	82.40%	86.44%	78.79%	0.8644
	BN-246	9	80.80%	76.27%	84.85%	0.8562
	HOA-112	17	87.20%	86.44%	87.88%	0.9081
	Bagging	40	88.80%	89.83%	87.88%	-
sMRI + fMRI	AAL-90	11	86.40%	84.75%	87.88%	0.8891
	AAL3-170	12	82.40%	79.66%	84.85%	0.8580
	BN-246	14	84.80%	83.05%	86.36%	0.8783
	HOA-112	22	88.00%	86.44%	89.39%	0.9124
	AAL-90+AAL3-170	8	87.20%	89.83%	84.85%	0.8903
	AAL-90+HOA-112	26	**92.00%**	**94.92%**	**89.39%**	**0.9502**
	BN-246+AAL3-170	20	86.40%	86.44%	86.36%	0.8914
	BN-246+HOA-112	29	88.00%	88.14%	87.88%	0.9135

## Data Availability

The datasets generated and analyzed during the present study are available from the corresponding author on reasonable request.

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
