# Peer review of "A Multi-Modal and Multi-Atlas Integrated Framework for Identification of Mild Cognitive Impairment"

_brainsci, 2022, doi:10.3390/brainsci12060751_

Round 1

Reviewer 1 Report

The paper is well written. There are no major spelling/grammar errors it describes the application of  fMRI and other similar methods it contains 5 figures and references.

the authors  recruited over 60 mild cognitive impairment patients and took healthy controls and control samples. they used brainnetome atlas

The field is novel and the described research is interesting. 

the conclusions are important, 

they used Tesla to analyze the data. 

The results 341 demonstrated obvious superiority of the proposed method compared to the single-mo-342 dality or single-atlas method, which can be used to improve the early diagnosis of MCI 343 and extended into the classification of other neurological and neuropsychiatric diseases.

Author Response

Thanks very much for your comments

Reviewer 2 Report

This manuscript has proposed a multi-atlas and multi-modal integrated framework to differentiate between MCI patients and HC subjects. Additionally, the authors have compared the differentiating capacity of the atlas used to extract cortical volume and functional parameters from structural and functional MRI and confirmed that using only optimal features showed higher accuracy than using full features. However, this study has several points to be improved.

1.     According to the MCI criteria described in this manuscript, it seems closer to subjective cognitive decline than MCI. It is necessary to describe the near-normal or normal performance of cognitive function described in (2) among the MCI criteria in detail based on the MMSE score and AVLT score (page3, lines 71-74).

2.     In the introduction, it is necessary to describe fractal-like dynamics in more detail to help readers understand them accurately (page 2, lines 37-38).

3.     Does "no visible vascular lesion" in the inclusion criteria mean that all subjects had a FAZEKAS score of 0? (page 3, lines 80-81).

4.     “we also tried applying the proposed classification method to full features without feature selection, and the accuracies were less than 70% in all kinds of single-modal or multi-modal data” (page 9, lines 270- 272). These results need to be presented in the supplemental material.

5.     It is necessary to provide evidence to state that HE abnormalities in thalamus and fusiform may provide insight into the mechanisms underlying MCI and AD patients (page 9, lines 304-306).

6.     The subjects of this study lacked information on amyloid-beta and tau deposition, and APOE genotype. Because these factors can affect brain structure and function, describing them in the Limitations section is also necessary.

Reviewer 3 Report

The authors proposed a multi-modality-multi-atlas classification framework combined with a feature selection procedure to achieve good classification accuracy that outperformed single-modality models on a modestly sized group of MCI and healthy controls. The most important features recapitulated some of the brain cortical regions implicated in previous studies.

While I am convinced by the superior performance of multimodal models over unimodal models, my primary concern about the manuscript is how multi-atlas was defined and explored. First, it was not very clear to me why the 2 data modalities and 4 atlases were chosen and combined in the current ways. Right now it seems that such choice is out of convenience (e.g., successfully utilized in past studies) without other considerations (e.g., using a structurally defined atlas for sMRI, a functionally defined atlas for fMRI). Second, this choice also confounds the examination of the role of atlas, i.e., there is no benchmark from a 'single atlas'  model (e.g., all modalities using the same atlas), so we cannot tell if using 2 different atlases does conveys any classification improvement. Third, the impact of multi-atlas seems to be less obvious than what the authors hypothesized:  the unimodal sMRI models based on different atlases (AAL and BN) yielded comparable accuracy, yet the authors still concluded that atlas choice mattered (p9). Finally, to be 'multi-altas', I would expect the authors to do something like pooling/bagging the classification results from several models using different atlases, or using more than one atlas for EACH modality in a single model (e.g., 2 atlases with very different spatial specificity for sMRI and another 2 for fMRI).

In my opinion, a more elaborated justification for the current design, or additional analyses, will be needed for further publication consideration.

Secondary comments

1) why is 30 chosen to be the max. number of features to be considered? Is it arbitrary?

2) why do the author prefer LOOCV over N-fold CV? Earlier literature proposed some shortcomings of LOOCV.

3) I am not particularly experienced with machine learning and I may be wrong, but I would expect the number of features to have clearer trends than what is shown in Figure 3. For example, there is not any obvious optimal number of features within each sMRI+fMRI models. Can the authors elaborate a bit more about the number of features in the manuscript?

4) I think some of the 'benchmark' analyses and results in the 2nd and 3rd paragraphs of Discussions should be mentioned and described in Methods and Results given their importance to the strength of the proposed method (impact of feature selection and linearity of kernels)

Minor points

1) P3. Is there a numerical definition/ cutoff of 'normal daily living activity'?

2) The writing can be improved, e.g., use of words (some of the 'could's are not very appropriate: p4 line 146 "the atlas is available"; not wrong but unusual use of plural forms like 'evidence' and 'divergence'); difficult to understand / incomplete sentences (p5 line 187 "As the top features were eliminated one by one, the classification process was repeatedly performed as the number of the remained features ranging from the first two to the full remained features.". Typos here and there e.g., P3 "Not meeting the dementia (?) according to DSM-IV" p9 line 298 it should be 'specificity' instead of 'specification'?

Round 2

Reviewer 3 Report

The authors have adequately answered my inquiries and the changes in the revised manuscript are well-done.